# TOAST: Transfer Learning via Top-Down Attention Steering

## Abstract

Transfer learning involves adapting a pre-trained model to novel downstream tasks. In this work, we empirically observe that current transfer learning methods often fail to focus on task-relevant features, potentially degrading transfer learning performance. We then explore refocusing model attention to improve transfer learning. We introduce **To**p-Down **A**ttention **St**eering (**TOAST**), a novel transfer learning algorithm that keeps the pre-trained backbone frozen, selects task-relevant features in the output, and feeds those features back to the model to steer the attention to the task-specific features. By refocusing the attention only, TOAST achieves state-of-the-art results on a number of transfer learning benchmarks, while having a small number of tunable parameters. Compared to fully fine-tuning, LoRA, and prompt tuning, TOAST substantially improves performance across a range of fine-grained visual classification datasets (*e.g.*, $82.6\% \rightarrow 86.2\%$ on FGVC). TOAST also outperforms the fully fine-tuned Alpaca and Vicuna models on instruction-following language generation.

## 1 Introduction

The prevailing approach for addressing novel tasks in deep learning is leveraging a pre-trained model and transferring it to the specific downstream task (Bommasani et al., 2021). Common approaches for transfer learning involve updating parts or all of the parameters in the model (*e.g.*, fine-tuning, LoRA (Hu et al., 2021)) or adding task-specific augmentation to the input (*e.g.*, prompt tuning (Lester et al., 2021), VPT (Jia et al., 2022)) in order to adjust model features for the downstream task.

However, we empirically find that previous transfer learning methods usually fail to focus the model's attention on task-relevant signals. For example, in Figure 1(b) we visualize the attention map of a ViT model pre-trained on ImageNet and transferred to downstream bird classification via fine-tuning, LoRA, or VPT. Such attention maps are often extremely noisy and fail to focus on task-specific objects. This encourages us to rethink the role of attention in transfer learning and if we can boost performance by refocusing the model's attention on task-related signals.

In this work, we show that *refocusing attention is key to transfer learning*. We introduce **To**p-Down **A**ttention **St**eering (**TOAST**), a novel transfer learning approach that learns new tasks by redirecting the attention to task-relevant features. This is achieved through a top-down attention module (Shi et al., 2023) which allows a model to adjust its attention in a task-adaptive way. The top-down attention module takes the output features from the backbone, selects the features that are relevant to the task, and then feeds them back into each self-attention layer in the backbone. These top-down signals will enhance the task-relevant features in each layer, and the feedforward backbone runs again with the enhanced feature, achieving stronger attention on the task-relevant signals. When transferring to different downstream tasks, TOAST simply freezes the pre-trained backbone and tunes the top-down attention module to steer the attention to task-specific signals (Figure 1(a)).

Remarkably, by simply refocusing attention, TOAST achieves state-of-the-art results on various transfer learning benchmarks. Compared to fully fine-tuning, LoRA, and VPT, TOAST significantly improves the performances on FGVC fine-grained classification (*e.g.*, $5\%$ improvement over fully fine-tuning on average accuracy), and obtains the best performance on 12 out of 19 tasks on VTAB benchmark (Zhai et al., 2019). Beyond visual recognition, TOAST can adapt large language models such as LLaMA (Touvron et al., 2023) for instruction-following language generation, resulting in more detailed and informed answers and outperforming fully fine-tuned Alpaca (Taori et al., 2023)

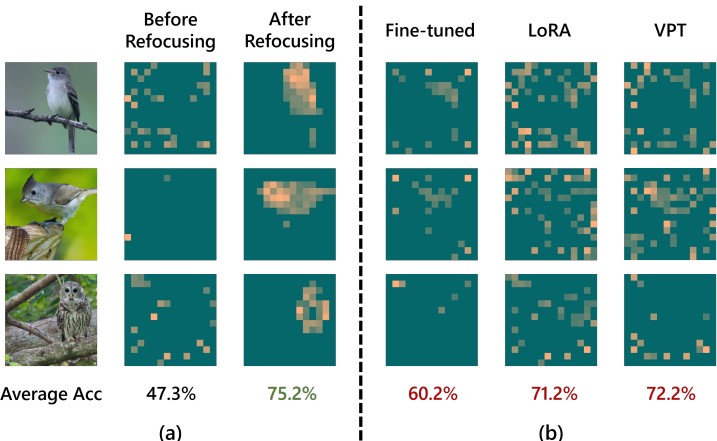

Figure 1: We take an ImageNet pre-trained ViT and transfer it to downstream bird classification using different transfer learning algorithms. Here we visualize the attention maps of these models. Each attention map is averaged across different heads in the last layer of ViT. **(a)** Our method, TOAST, is able to refocus the attention of a pre-trained backbone onto task-specific features, improving the downstream performance by a large margin. **(b)** Previous transfer learning methods such as fine-tuning, LoRA, and VPT fail to focus on task-relevant objects, achieving suboptimal performance.

and Vicuna (Chiang et al., 2023). We also explore variants of TOAST that have improved parameter efficiency and computational efficiency. These observations strengthen our idea that refocusing attention is key to transfer learning and sheds light on future exploration in the field.

**Contributions. 1)** We empirically observe that previous transfer learning methods struggles at focusing attention on task-relevant objects on downstream tasks. **2)** We propose TOAST, an algorithm that performs transfer learning by steering model attention on the task-relevant objects. **3)** We show that by simply refocusing attention, TOAST is able to significantly improve transfer learning performance on several vision and language benchmarks.

## 2 RELATED WORK

**Transfer learning** refers to adapting a pre-trained model to a downstream task, which has become the paradigm of tackling unseen tasks for both vision (Zhuang et al., 2020) and language (Devlin et al., 2018; Radford et al., 2018). Normal approaches for transfer learning involve tuning all the parameters (*i.e.*, fully fine-tuning) or part of the parameters (*e.g.*, only tuning the last few layers (Yosinski et al., 2014) or the bias terms (Cai et al., 2020) of the network). Recent progress on large foundation models (Bommasani et al., 2021; Dehghani et al., 2023; Touvron et al., 2023) has promoted the exploration of Parameter-Efficient Fine-Tuning (PEFT) which is able to adapt the model by tuning only a small number of parameters (usually less than 1% of all the parameters) and thus is more suitable for large models with billions of parameters. Common strategies include freezing the pre-trained backbone and adding additional tunable parameters (*e.g.*, Adapter (Houlsby et al., 2019), LoRA (Hu et al., 2021)) or task-specific input augmentation (*e.g.*, prefix tuning (Liu et al., 2021), prompt tuning (Lester et al., 2021; Jia et al., 2022)). However, either fine-tuning or PEFT methods suffer from focusing attention on downstream tasks as shown in this paper, and we show TOAST can significantly improve downstream performances by refocusing the attention.

**Top-down attention and its relation to transfer learning**. Top-down attention, one of the hallmarks of the human visual system, is the ability to selectively focus one's attention on the input signals that are relative to the current task or goal (Carrasco, 2011; Zhaoping, 2014). Top-down attention has been applied to different computer vision tasks such as object detection (Oliva et al., 2003), image captioning (Xu et al., 2015), and visual question answering (Anderson et al., 2018; Xu & Saenko, 2016). Recent work (Shi et al., 2023) has designed a top-down attention module for Transformer, which we adopt in this work. On the other hand, previous studies on human perceptual learning have indicated a close relationship between top-down attention and how humans adapt to unseen

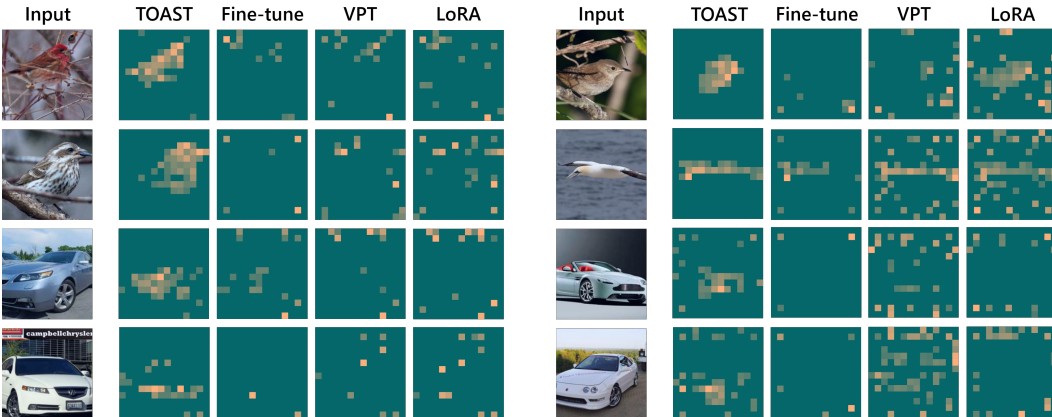

Figure 2: Comparison between the attention map of different models. The first two rows are evaluated on bird classification, and the last two on car classification. The attention of fine-tuning, LoRA, and VPT is noisy, while TOAST has cleaner attention that is focused on the task-relevant signals such as the foreground birds or the headlights and the badge of the cars.

tasks. Specifically, top-down attention facilitates learning new tasks by extracting task-relevant features while ignoring the distracting information (Lavie, 1995; Rees et al., 1997). Additionally, it is shown that only the task-relevant features are enhanced during adaptation, while the irrelevant features remain undistorted (Ahissar & Hochstein, 1997; Schoups et al., 2001; Tsushima & Watanabe, 2009). This stands in contrast with transfer learning algorithms such as fully fine-tuning where all the pre-trained features are modified, indicating that the key to learning new tasks is adjusting the attention, not the pre-trained features.

## 3  Previous Transfer Learning Methods Fail to Focus Attention

In this section we make the empirical observation that, when transferring a pre-trained model to downstream tasks using previous transfer learning methods (*e.g.*, fully fine-tuning, VPT (Jia et al., 2022), LoRA (Hu et al., 2021)), the model usually fails to focus its attention on objects or features that are relevant to the downstream task.

The experimental setup follows previous literature. Specifically, we evaluate on FGVC which contains 5 downstream visual classification tasks. We take an ImageNet pre-trained ViT (Dosovitskiy et al., 2020) and tune the model on the downstream tasks using fully-finetuning, LoRA, or VPT. Then we take the attention map from the [CLS] token to other tokens in the last layer of ViT. The attention map is averaged across different heads. Results are visualized in Figure 2. Examples come from downstream bird or car classification. we can see that three transfer learning methods all show extremely noisy attention maps and do not focus the attention on the task-relevant objects. To quantitatively evaluate the attention, we measure the mean IoU between the attention map and the foreground segmentation mask on FGVC bird classification dataset (CUB). As shown in Table 1, all three methods admit low segmentation results, indicating the misalignment between attention and the foreground objects.

Table 1: The mean IoU between attention map of each transfer learning method with the foreground object mask.

| Method | mIoU@CUB |
|---|---|
| Fine-tune | 5.74 |
| LoRA | 21.49 |
| VPT | 16.3 |
| TOAST | **41.28** |

These observations indicate, surprisingly, that the prevailing transfer learning methods all suffer on focusing attention on the task-relevant objects. This inspire us to explore whether refocusing the attention back to the objects can improve the transfer learning performance. In the next section, we propose an algorithm that performs transfer learning by steering and refocusing attention, and we show in Section 5 that attention refocusing can indeed significantly improve transfer learning performance.

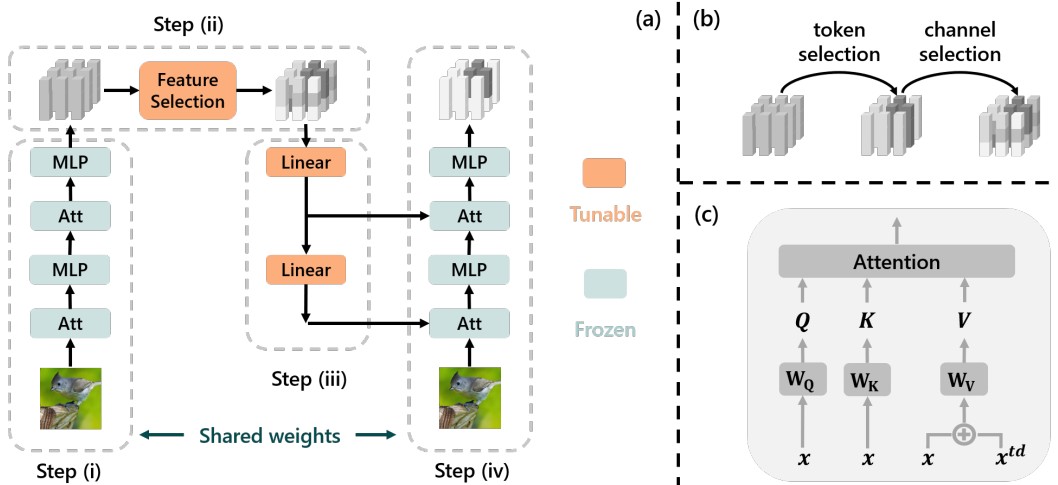

Figure 3: **(a)** Overview of TOAST. In addition to the regular feedforward transformer which contains interleaving MLP and Attention blocks, we add a feature selection module and a feedback path consisting of linear layers. Inference has four steps: (i) the input goes through the feedforward transformer, (ii) the output tokens are softly reweighted by the feature selection module based on their relevance to the task, (iii) the reweighted tokens are sent back through the feedback path, and (iv) we run the feedforward pass again but with each attention layer receiving additional top-down inputs. During the transfer, we only tune the features selection module and the feedback path and keep the feedforward backbone frozen. **(b)** The feature selection module first selects the task-relevant tokens by reweighting the tokens based on their similarity to the task embedding, then selects the task-relevant channels by applying a task-specific linear transform on the channel dimension. **(c)** In the second feedforward pass, each self-attention layer receives an additional top-down input, which is added on the value matrix.

## 4 TOP-DOWN ATTENTION STEERING

We propose **To**p-**D**own **A**ttention **St**eering (**TOAST**), an algorithm of transfer learning via attention steering. Specifically, TOAST first arms the pre-trained model with a top-down attention module, which is then tuned to steer model attention during transfer learning. We first introduce the top-down attention module in Section 4.1, and then describe the attention steering pipeline in Section 4.2.

### 4.1 TRANSFORMER WITH TOP-DOWN ATTENTION

Transformer model is usually bottom-up, *i.e.*, its attention only depends on the input, and as a consequence, it normally highlights all the salient features in the input signal. As opposed to bottom-up attention, top-down attention endows the ability to adapt one's attention according to the high-level goal or task and only focuses on the task-relevant objects (Carrasco, 2011; Zhaoping, 2014).

In this work, we follow the top-down attention design proposed in (Shi et al., 2023), which is illustrated in Figure 3(a). Specifically, for a regular transformer which is purely feedforward, we add a feature selection module and a feedback path for top-down attention. Inference of the network contains four steps: **(i)** pass the input through the feedforward path to get an initial output, **(ii)** select which features in the output is useful for the current task, **(iii)** the selected features are passed through the feedback path and sent back to each self-attention module, and **(iv)** run the feedforward pass again but with each self-attention receiving the top-down signal as additional input. In this way, the task-relevant information is enhanced in each layer, achieving top-down attention.

Within the network, the feedforward path is a regular transformer, and the rest is described below:

**Feature selection (Step (ii))**. From the output of the feedforward backbone, this module selects the features that are useful for the task at hand. This includes the selection of both the tokens and

the channels that are task-relevant. Figure 3(b) illustrate the process. Specifically, denoting the output from the first feedforward pass by $(\mathbf{z}_i)_{i=1}^N$ where $\mathbf{z}_i \in \mathbb{R}^d$ is the $i$-th output token, the feature selection operates on each token and outputs $\widetilde{\mathbf{z}}_i = \mathbf{P} \cdot sim(\mathbf{z}_i, \xi) \cdot \mathbf{z}_i$, where $\xi \in \mathbb{R}^d$ and $\mathbf{P} \in \mathbb{R}^{d \times d}$ are task-specific parameters, and $sim(\cdot, \cdot)$ is cosine similarity clamped to $[0, 1]$. Here $\xi$ acts as a task embedding that encodes what kind of tokens are important for the task, and each token $\mathbf{z}_i$ is reweighted by its relevance (measured by cosine similarity) with the task embedding, simulating the token selection. Then the linear transform by $\mathbf{P}$ executes the channel selection for each token.

**Feedback path (Step (iii))**. After feature selection, the output tokens are sent back through the feedback path. The feedback path contains the same number of layers as the feedforward path, and each layer is a simple linear transform. The output from each layer goes through another linear transform and is sent into the self-attention module as the top-down input for the second feedforward.

**Self-attention with top-down input (Step (iv))**. In the second feedforward pass, each self-attention module receives an additional top-down input. As shown in Figure 3(c), we simply add it to the value matrix while keeping the query and key untouched, *i.e.*, $\mathbf{Q}, \mathbf{K}, \mathbf{V} = \mathbf{W}_Q \mathbf{X}, \mathbf{W}_K \mathbf{X}, \mathbf{W}_V (\mathbf{X} + \mathbf{X}^{td})$, where $\mathbf{X}$ is the regular bottom-up input to the self-attention module, and $\mathbf{X}^{td}$ is the top-down input. Then the regular self-attention on $\mathbf{Q}, \mathbf{K}, \mathbf{V}$ follows.

## 4.2 TOP-DOWN ATTENTION STEERING

Given a pre-trained transformer, TOAST randomly initialize a top-down attention module and follows a two-stage pipeline: **(i)** pre-tuning the top-down attention on a general public dataset (*e.g.*, ImageNet (Deng et al., 2009) for vision or OpenWebText (Gokaslan & Cohen, 2019) for language) to get a better initialization, and **(ii)** tuning the top-down attention on the downstream task. In both stages, we freeze the pre-trained backbone and only tune the top-down attention module (Figure 3(a)).

**Pre-tuning stage**. Since the top-down attention module is randomly initialized, directly tuning it on downstream tasks might lead to suboptimal performance (see ablation in Section 5.6). To this end, we propose to first pre-tune the top-down attention on a general public dataset such as ImageNet or OpenWebText to get a better initialization. During pre-tuning, except for the regular supervised or unsupervised loss, we also add the variational loss proposed in (Shi et al., 2023), which encourages the feedback path to reconstruct the input from the output, as a regularization on the feedback weights.

**Tuning stage**. When transferring to the downstream task, TOAST only fine-tunes the parameters in the top-down attention module. In this case, around $15\%$ of the parameters are updated for vision models such as ViT, and around $8\%$ of the parameters are tuned for large language models such as LLaMA. We notice that most of the tunable parameters are from the feedback layers, each of which contains a $d \times d$ matrix and is large when the feature dimension $d$ is high. To further promote parameter efficiency, we also propose TOAST-Lite, which applies LoRA on the feedback layers. In this way, only less than $1\%$ of the parameters are tuned. We empirically show that TOAST-Lite performs on par with TOAST on certain tasks while slightly worse on others (see Section 5.4).

## 5 EXPERIMENTS

In this section, we first evaluate the attention steering process in TOAST (Section 5.1). Then we evaluate TOAST's performance on visual classification and segmentation (Section 5.2) and language generation (Section 5.3). We also explore parameter-efficient (Section 5.4) and computation-efficient (Section 5.5) version of TOAST. Finally, we conduct ablation studies on the designing choices of TOAST (Section 5.6). Please refer to Appendix for additiona results such as TOAST adapted ConvNet architectures.

**Datasets**. We pre-tune the top-down attention on ImageNet (Deng et al., 2009) for vision models and a subset of OpenWebText (Gokaslan & Cohen, 2019) for language models. For evaluation on visual classification, we follow the protocols in (Jia et al., 2022) and evaluate on FGVC and VTAB-1k (Zhai et al., 2019). FGVC contains 5 datasets of fine-grained natural image classification, each with around 10k training images. VTAB has 19 classification tasks that span natural image classification, specialized image classification (satellite, medical, *etc.*), and image structure understanding (*e.g.*,

object counting, depth estimation), with each task containing 1k training images. For evaluation on language generation, we compare to Stanford Alpaca (Taori et al., 2023) by training on the same Alpaca dataset which contains 52k instruction-following data, and compare to Vicuna (Chiang et al., 2023) by training on an open-source dataset collected from ShareGPT conversations.

**Experimental setup**. We compare with several baselines for transfer learning: (i) **Linear** freezes the pre-trained backbone and only tunes a linear head on top of it, (ii) **Fully fine-tuning** tunes the whole backbone, (iii) **VPT** (Jia et al., 2022) adds additional prompt tokens into the input as well as each intermediate layer and only tunes the prompt tokens, (iv) **LoRA** (Hu et al., 2021) adds low-rank matrices onto the linear transform weights in the network and only tunes the low-rank matrices. The pre-trained backbone for visual classification is by default ViT-B (Dosovitskiy et al., 2020) pre-trained on ImageNet-1k. To align with the literature, we also test the performance on VTAB-1k using a ViT-B pre-trained on ImageNet-21k. For language generation, we use LLaMA-7B and LLaMA-13B (Touvron et al., 2023) as the pre-trained backbones.

## 5.1 UNDERSTANDING AND EVALUATING ATTENTION STEERING IN TOAST

To understand how TOAST adapts to downstream tasks by refocusing its attention, we visualize how the attention changes during the inference of the top-down attention model (see Section 4.1). In Figure 4, we show the attention map in the first feedforward pass, the similarity map in the feature selection step, as well as the attention in the second feedforward pass. We take FGVC bird classification as our example. One can see that in the first feedforward the pre-trained model fails to concentrate on the task-relevant objects. TOAST addresses this problem with two stages. First, it selects the task-relevant features with the feature selection module. We can observe from the cosine similarity map that it coarsely selects the task-relevant objects. Then the reweighted tokens are sent back to the network to enhance the task-relevant features in the second feedforward run. We can see the attention is refined and refocused on the bird, providing better representations for the downstream task.

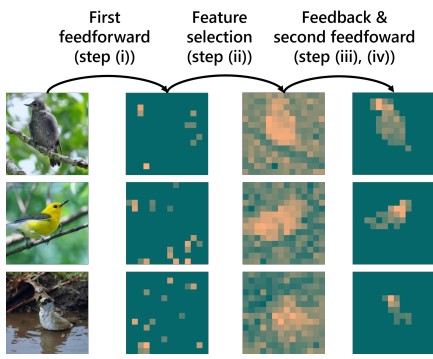

Figure 4: Visualization of the attention maps during each step of model inference. The attention is extremely noisy in the first feedforward. The feature selection step coarsely selects the task-relevant features, and in the second feedforward, the attention is refined and refocused on the task-relevant objects.

We compare attention maps between TOAST and other baselines. We use examples from FGVC bird and car classification. As shown in Figure 2, for bird classification, TOAST clearly focuses on the foreground birds while other methods either have noisy attention or completely ignore the foreground object. On car classification, we observe that TOAST tends to concentrate on the headlights and the badge of the car which helps discriminate different brands of cars, while other methods have less explainable and noisier attention. For a quantitative comparison, Table 1 shows attention map of TOAST has nearly 2x mIoU than baselines, indicating its attention is much more aligned with foreground objects.

## 5.2 EVALUATION ON VISUAL CLASSIFICATION AND SEGMENTATION

We first evaluate TOAST on FGVC which contains 5 datasets of fine-grained natural image classification (Table 2). We can see that TOAST outperforms fully fine-tuning as well as other baselines by a large margin. Especially, TOAST improves the average accuracy by 5% over fine-tuning while training less parameters. This is possibly due to the over-fitting issue in fine-tuning, which is also observed in VTAB-1k experiments. LoRA and VPT alleviate the over-fitting issue by only tuning a small part of the parameters, and reach higher downstream performance. TOAST achieves the highest avarage accuracy by refocusing the model's attention to objects relevant to downstream tasks.

We also test on VTAB-1K, which contains 19 datasets of natural and specializedimage classification as well as image structure understanding. Since each dataset only has 1k training images, VTAB simulates a setting with higher data scarcity. Results are shown in Table 3. Overall, TOAST reaches

Table 2: Results on FGVC fine-grained classification. TOAST is able to outperform previous baselines by a large margin on different tasks and achieves state-of-the-art average performance.

|  | CUB | Bird | Flower | Dog | Car | *Avg* |
|---|---|---|---|---|---|---|
| Linear | 76.8 | 47.3 | 81.7 | **97.7** | 60.3 | *72.8* |
| Fine-tune | 80.5 | 60.2 | 86.9 | 94.7 | 83.2 | *81.1* |
| VPT | 82.1 | 72.1 | 83.6 | 97.3 | 77.8 | *82.6* |
| LoRA | 82.5 | 71.2 | 81.2 | 97.5 | 76.6 | *81.8* |
| TOAST | **85.0** | **75.2** | **88.7** | 97.4 | **84.9** | ***86.2*** |

Table 3: Results on VTAB-1K benchmark. TOAST outperforms previous baselines on 11 out of 18 tasks for ImageNet-1k pre-trained model and 10 out of 18 tasks for ImageNet-21k pre-trained model. All methods are implemented in the same environment.

| | Natural | | | | | | | Specialized | | | | Structured | | | | | | | | |
|---|---|---|---|---|---|---|---|---|---|---|---|---|---|---|---|---|---|---|---|---|
| | Cifar100 | Caltech101 | DTD | Flower102 | Pets | SVHN | Sun397 | Camelyon | EuroSAT | Resisc45 | Retinopathy | Clevr-Count | Clevr-Dist | DMLab | KITTI-Dist | dSpr-Loc | dSpr-Ori | sNORB-Azim | sNORB-Ele | Average |
| *ImageNet-1k pre-trained* | | | | | | | | | | | | | | | | | | | | |
| Fine-tune | 44.7 | 77.3 | 55.5 | 74.5 | 86.0 | **85.1** | 17.4 | **84.9** | 95.0 | 82.8 | 74.2 | 60.2 | 53.1 | 33.5 | 77.6 | 61.9 | 39.0 | 15.0 | 36.6 | 64.7 |
| VPT | 65.3 | 90.5 | 67.7 | 88.3 | 88.6 | 82.2 | 40.6 | 82.3 | 94.5 | 83.1 | 74.0 | 51.5 | 51.1 | 44.1 | 69.3 | 63.8 | 49.5 | 25.3 | 28.6 | 68.7 |
| LoRA | 69.3 | 88.8 | 66.6 | 90.3 | **90.3** | 81.9 | 41.5 | 83.4 | 94.8 | 83.5 | **75.0** | **66.8** | 56.9 | **48.9** | 77.6 | 76.2 | **53.5** | 26.6 | 37.1 | 71.7 |
| TOAST | **73.8** | **92.1** | **68.7** | **93.0** | 89.0 | 76.3 | **41.9** | 82.8 | **95.3** | **85.7** | 74.6 | 61.2 | **58.7** | 43.5 | **78.8** | **86.1** | 51.2 | **27.0** | **43.4** | **72.5** |
| *ImageNet-21k pre-trained* | | | | | | | | | | | | | | | | | | | | |
| Fine-tune | 70.2 | 85.8 | 64.3 | 97.5 | 85.8 | **85.9** | 40.0 | 78.2 | 95.7 | 83.8 | 73.9 | 53.1 | 57.3 | 37.5 | 68.2 | 60.5 | 35.2 | 18.8 | 28.0 | 67.8 |
| VPT | 75.4 | 88.7 | 66.3 | 98.1 | 87.3 | 73.7 | 52.3 | 80.3 | 93.5 | 83.4 | 74.1 | 49.6 | 58.1 | 41.9 | 62.7 | 65.1 | 42.9 | 24.0 | 24.2 | 68.8 |
| LoRA | **83.6** | 89.4 | 66.2 | 98.6 | 89.4 | 83.8 | 52.6 | 81.1 | **95.8** | 84.6 | **74.7** | **77.6** | 59.5 | **46.8** | 74.1 | 73.0 | **48.6** | **25.6** | 32.2 | **73.1** |
| TOAST | 82.1 | **90.5** | **70.5** | **98.7** | **89.7** | 71.9 | **53.3** | 84.3 | 95.5 | **85.5** | 74.2 | 75.4 | **60.8** | 44.7 | **77.5** | 73.9 | 47.5 | 24.5 | **33.7** | **73.1** |

competitive performances, outperforming other baselines in 12 of 19 datasets for ImageNet-1k pre-trained model and 11 of 19 datasets for ImageNet-21k pre-trained model. We notice that TOAST shows more advantages on natural image classification than on specialized image classification or image structure understanding. This is possibly because the last two task categories have too large gaps from the pre-training task of natural image classification, and the features relevant to downstream tasks are absent in the pre-trained backbone. In this case, only refocusing the attention is not enough and tuning the feedforward backbone is essential for learning the features.

Beyond visual recognition, previous work (Jia et al., 2022) shows that PEFT methods are not comparable to fine-tuning on dense prediction tasks such as semantic segmentation. Here we test TOAST on semantic segmentation on two datasets, PASCAL VOC (Everingham et al.) and ADE20K (Zhou et al., 2017). We use ImageNet-21k pre-trained ViT-B as the backbone and UperNet (Xiao et al., 2018) as the segmentation head. From Table 4, we observe that TOAST has better performance than VPT and LoRA, although still underperforms fine-tuning. One possible reason is that the backbone is pre-trained on image classification which has too large a discrepancy with segmentation tasks in terms of the hierarchy and semantics of the required visual representations.

Table 4: Results on Semantic Segmentation. TOAST consistently outperforms LoRA and VPT but still lags behind fully fine-tuning.

|  | PASCAL VOC | ADE20K |
|---|---|---|
| Fine-tune | **82.05** | **47.89** |
| VPT | 76.80 | 41.42 |
| LoRA | 78.43 | 42.94 |
| TOAST | 80.44 | 45.11 |

## 5.3 EVALUATION ON LANGUAGE GENERATION

We apply TOAST to LLaMA-7B and LLaMA-13B (Touvron et al., 2023) and tune it on the Alpaca dataset (Taori et al., 2023) to compare to Stanford Alpaca, and on the open-source ShareGPT dataset to compare to Vicuna (Chiang et al., 2023). For evaluation, we follow the pipeline in Vicuna (Chiang

Table 5: Results of Instruction Tuning. We use GPT-4 to evaluate the performance. LoRA has inferior performance while TOAST is able to outperform fully fine-tuned Alpaca and Vicuna. [†]We train the Alpaca and Vicuna baselines using the same open-source data as LoRA and TOAST and report the performance thereof.

|  | Generic | Knowledge | Roleplay | Common | Fermi | Counterfact | Writing | *Avg* |
|---|---|---|---|---|---|---|---|---|
| Alpaca-7B[†] | 7.8 | 8.5 | **8.0** | **8.0** | 4.3 | **8.7** | **9.7** | *7.9* |
| - LoRA | 6.7 | 7.3 | 6.7 | 7.0 | 5.0 | 7.3 | 8.0 | *6.9* |
| - TOAST | **8.0** | **9.0** | 7.7 | **8.0** | **7.0** | 8.0 | 8.7 | ***8.1*** |
| Vicuna-7B[†] | 8.3 | 8.8 | 8.2 | 8.0 | **6.7** | **7.7** | 8.8 | *8.1* |
| - TOAST | **8.7** | **9.0** | **8.7** | **9.0** | 6.5 | 7.0 | **9.0** | ***8.3*** |
| Vicuna-13B[†] | 7.6 | 8.5 | **9.3** | 8.2 | **7.0** | **8.0** | 8.7 | *8.2* |
| - TOAST | **8.9** | **9.0** | 8.0 | **9.0** | 6.7 | **8.0** | **9.0** | ***8.4*** |

et al., 2023), *i.e.*, we ask questions that span different categories and use GPT-4 (OpenAI, 2023) to score the answers provided by the model. Scores are based on the answers' helpfulness, relevance, accuracy, and level of detail. In Table 5, we compare the scores of different models on each category of questions. All the scores are on a scale of 1-10. Overall, TOAST has better performance than fine-tuning for each baseline model, outperforming the fully-finetuned Alpaca-7B, Vicuna-7B and 13B, while tuning only $8\%$ of the parameters. It is worth noting that Vicuna-7B with TOAST can already outperform the fine-tuned Vicuna-13B while having only half of the parameters. On the other hand, LoRA is less competitive and lags far behind the fine-tuning baseline on Alpaca. Looking into the output response from TOAST, we observe that TOAST is especially better at providing detailed and informed answers, which makes it a better assistant for knowledge-based questions (*e.g.*, Generic, Knowledge, Common). Please refer to Appendix for examples.

## 5.4 IMPROVING PARAMETER-EFFICIENTY OF TOAST

Although TOAST tunes only a small fraction of the parameters (*e.g.*, $8\%$ of parameters for LLaMA), the number of tunable parameters is still large especially for LLMs which have billions of parameters. We observe that most of the tunable parameters are from the feedback path. To further improve parameter efficiency, we propose TOAST-Lite which applies LoRA on the feedback path. In this way, only less than $1\%$ of the parameters are tuned. Here we evaluate the performance of TOAST-Lite on FGVC and Alpaca. Results are shown in Table 6. We can see that although TOAST-Lite tunes much fewer parameters than TOAST, it performs on par with TOAST on FGVC. It also largely outperforms

Table 6: Evaluation of TOAST-Lite on FGVC visual classification and Alpaca language generation. TOAST-Lite outperforms LoRA and VPT with a similar number of tunable parameters.

|  | FGVC | | Alpaca | |
|---|---|---|---|---|
|  | #Param | Acc | #Param | Score |
| Fine-tune | 87M | 81.1 | 7B | 7.9 |
| LoRA | 0.3M | 79.8 | 4.2M | 6.9 |
| VPT | 0.9M | 78.0 | - | - |
| TOAST | 14M | **86.2** | 537M | **8.1** |
| TOAST-Lite | 0.9M | 86.0 | 19M | 7.4 |

LoRA and VPT while having a similar level of parameter efficiency. For Alpaca, TOAST-Lite has a degraded performance compared to TOAST but still outperforms LoRA, making it a strong baseline for Parameter-Efficient Fine-Tuning. See Appendix for more results of TOAST-Lite.

## 5.5 IMPROVING COMPUTATION-EFFICIENCY OF TOAST

Behind the promising performances, one drawback of TOAST is the computation overhead since the feedforward path is run twice, which approximately doubles the FLOPs of the model. To improve computational efficiency, we propose TOAST-Late, a variant of TOAST that only runs the feedback path for the later stage of the backbone, *i.e.*, the feedback signal from the last layer is sent to the middle layer instead of the first layer, and the early layers do not receive any feedback signals. In this way, we can share the outputs of the early layers across two feedforward runs without rerunning this part. In Table 7, we compare the computational efficiency of TOAST, TOAST-Late, as well as the

Table 7: Results of TOAST-Late on FGVC visual classification and Vicuna language generation. The default backbones are ViT-B or LLaMA-7B. † indicates using larger backbone (ViT-L or LLaMA-13B) for transfer learning. Although TOAST doubles the FLOPs over fine-tuning, it outperforms the larger fine-tuned models while having fewer FLOPs or shorter training time. TOAST-Late reduces the FLOPs but with a cost of degrading performances compared to TOAST.

| | FGVC | | | Vicuna | | |
| | Train Time (↓) | FLOPs (↓) | Acc | Train Time (↓) | FLOPs (↓) | Score |
|---|---|---|---|---|---|---|
| Fine-tune | 31.7 | 1x | 81.1 | 60.7 | 1x | 8.1 |
| - Larger Model† | 53.6 | 3.5x | 86.0 | 103.2 | 1.9x | 8.2 |
| TOAST | 29.4 | 2.1x | **86.2** | 32.7 | 2.1x | **8.3** |
| TOAST-Late | 25.0 | 1.4x | 84.1 | 26.1 | 1.4x | 8.2 |

fine-tuning baselines. Here we report both training-time efficiency (measured by the wall clock train time of each epoch or iteration) and inference-time efficiency (measured by FLOPs). We first see that TOAST has 2.1x FLOPs compared to fine-tuning baseline. However, it is worth noting that TOAST actually outperforms the larger fine-tuned models while having fewer FLOPs or shorter training time. On the other hand, TOAST-Late reduces the FLOPs from 2.1x to 1.4x, while still outperforming the fine-tuning baseline, although at the slight cost of degraded performance compared to TOAST. We also observe that both TOAST and TOAST-Late reduces the training time compared to fine-tuning. This is because the feedforward backbone is frozen, saving the time of building the computational graph in the first feedforward run, and a much smaller number of tunable parameters greatly saves the cost of updating parameter gradients.

## 5.6 ABLATION STUDIES

We conduct ablation studies to show the importance of several designs of TOAST: (i) the pre-tuning stage which provides a better initialization of the top-down attention module, and (ii) the token-wise and channel-wise attention in the top-down attention module. For each ablation, we remove the pre-tuning stage, remove the token selection in the feature selection module, and freeze the channel selection as well as the feedback path, respectively. Note that we freeze the feedback path because it contains linear transforms on the channel dimension and thus also plays a role in channel selection. Results are shown in Table 8.

Table 8: Ablation studies on the pre-tuning stage, the token-wise and channel-wise attention in TOAST.

| Model | FGVC Avg Acc |
|---|---|
| TOAST | **86.2** |
| w/o pre-tuning | 81.9 |
| w/o token att | 82.8 |
| w/o channel att | 74.7 |

First, we observe that TOAST without pre-tuning has a considerable performance drop from 86.2% to 81.9%. This indicates a proper initialization of the top-down attention module is crucial. Notably, TOAST without pre-tuning still outperforms fine-tuning, proving the effectiveness of attention refocusing. Second, we can see that removing the token-wise attention or channel-wise attention will both harm the performance. Specifically, removing channel-wise attention has a larger impact, indicating that at the same position in an image, the pre-trained model is usually not focusing on the features concerned by downstream tasks.

## 6 CONCLUSION

This work is motivated by the empirical observation that previous transfer learning methods often fail to focus the model's attention on task-relevant signals, which possibly leads to suboptimal performance on downstream tasks. We show that refocusing attention is the key to better transfer learning performance. We propose Top-Down Attention Steering (TOAST) which transfers to a new task by steering the attention onto the task-specific features. Specifically, TOAST freezes the pre-trained backbone and tunes an additional top-down attention module on the downstream task to steer the attention. Compared to previous baselines, TOAST is able to achieve state-of-the-art results on fine-grained visual classification as well as instruction-following language generation while only tuning a small portion of the parameters.

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

# A ADDITIONAL IMPLEMENTATION DETAILS

## A.1 PRE-TRAINED BACKBONE

For the feedforward ViT-B backbone with ImageNet-1k pre-training, we use the implementation from DeiT Touvron et al. (2021) and pre-train on ImageNet-1k with the same recipe, *i.e.*, using AdamW optimizer to pre-train for 300 epochs, with a batch size of 512, a base learning rate of 5e-4, and 5 warm-up epochs. For the ViT-B and ViT-L backbone with ImageNet-21k pre-training, we take the checkpoints from HuggingFace[1] [2] and convert them into DeiT style. For ConvNeXt we directly borrow the implementation and checkpoints from the original GitHub repository[3]. For LLaMA-7B and LLaMA-13B, we take the checkpoints provided by the community[4] [5].

## A.2 PRE-TUNING STAGE

For vision models such as ViT and ConvNeXt, we first add a randomly initialized top-down attention module onto the pre-trained backbone and then pre-tune the top-down attention module on ImageNet-1k classification. In this process, the feedforward backbone is frozen. We pre-tune the model for 30 epochs using the AdamW optimizer, with 3 warm-up epochs, and 3 cool-down epochs, a learning rate of 0.0005. We also disable the cutmix and mixup. Except for the supervised loss, we also add the variational loss Shi et al. (2023) which encourages the feedback layer in $\ell$-th layer to reconstruct the input feature to $\ell$-th layer from its output. We set the weight of variational loss as 0.03.

For language models, we pre-tune on a subset of OpenWebText Gokaslan & Cohen (2019). The subset contains 200k lines sampled from the original dataset. We train for 1 epoch with a batch size of 32 and 4 gradient accumulation steps. We use a learning rate of 3e-5. We use DeepSpeed[6] parameter offloading to avoid OOM errors.

## A.3 TUNING STAGE

For FGVC experiments, we use the training recipe in Jia et al. (2022). Specifically, on each dataset, we use a learning rate of 0.01 for TOAST, LoRA, and VPT, and use 0.003 for fine-tuning. We use a batch size of 32.

For VTAB experiments, we follow Jia et al. (2022) to do a grid search on the best learning rate and weight decay for each model and each dataset. Specifically, we take 800 images from the training set to train the model and use the rest 200 images for validation. We pick the set of hyperparameters that has the highest validation performance. Then we use the same hyperparameters to train the model on all 1000 images and test on the testing set. For each dataset, we run it five times with random seeds and report the average results.

For experiments on Alpaca and Vicuna, we use the same training recipe as the Stanford Alpaca repository[7]. During the evaluation, we use a temperature of 0.7. The evaluation protocol follows

---

[1]https://huggingface.co/google/vit-base-patch16-224-in21k

[2]https://huggingface.co/google/vit-large-patch16-224-in21k

[3]https://github.com/facebookresearch/ConvNeXt

[4]https://huggingface.co/decapoda-research/llama-7b-hf

[5]https://huggingface.co/decapoda-research/llama-13b-hf

[6]https://github.com/microsoft/DeepSpeed

[7]https://github.com/tatsu-lab/stanford_alpaca

Table 9: Results on FGVC with ConvNeXt-B backbone.

|  | CUB | Birds | Flower | Dogs | Cars | *Avg* |
|---|---|---|---|---|---|---|
| Fine-tune | 87.5 | 72.3 | 97.1 | 86.3 | **87.7** | *86.2* |
| LoRA | 89.6 | 75.8 | **99.3** | **88.5** | 67.7 | *84.2* |
| TOAST | **90.2** | **85.6** | 99.2 | 88.4 | 85.8 | *89.8* |

Table 10: Results on FGVC with ViT-L backbone pre-trained on ImageNet-21k.

|  | CUB | Bird | Flower | Dog | Car | *Avg* |
|---|---|---|---|---|---|---|
| Fine-tune | 88.3 | 69.4 | 98.1 | 89.8 | 84.3 | 86.0 |
| LoRA | 89.1 | 73.9 | 98.2 | **94.3** | 78.1 | 86.7 |
| TOAST | **89.5** | **75.4** | **98.5** | 93.4 | **85.3** | *88.4* |

Table 11: Results on FGVC with TOAST and TOAST-Lite. TOAST-Lite is able to improve the performance by a large margin over LoRA and VPT while tuning a similar number of parameters.

|  | # Params | CUB | Birds | Flower | Dogs | Cars | *Avg* |
|---|---|---|---|---|---|---|---|
| Linear | 0.2M | 76.8 | 47.3 | 81.7 | **97.7** | 60.3 | *72.8* |
| Fine-tune | 87M | 80.5 | 60.2 | 86.9 | 94.7 | 83.2 | *81.1* |
| VPT | 0.9M | 82.1 | 72.1 | 83.6 | 97.3 | 77.8 | *82.6* |
| LoRA | 0.3M | 82.5 | 71.2 | 81.2 | 97.5 | 76.6 | *81.8* |
| TOAST | 14M | **85.0** | 75.2 | 88.7 | **97.4** | **84.5** | *86.2* |
| TOAST-Lite | 0.9M | 84.5 | **76.9** | **89.4** | **97.4** | 82.0 | *86.0* |

the one in Vicuna Chiang et al. (2023) except we sample 30 questions from the original list of 80 questions.

## B  ADDITIONAL RESULTS ON TOAST ADAPTED TO DIFFERENT MODEL ARCHITECTURES

**TOAST is adaptable to Convnets**. In previous experiments, we use Transformer as the backbone. We show that we can also use TOAST on convolutional networks (Convnets). First, we need to design a top-down attention module for Convnets: (i) we keep the design of the feature selection module, (ii) we change the linear layers in the feedback path into deconvolutional layers so that the bottom-up and top-down signals in each layer have the same shape, (iii) since there is no self-attention in Convnets, we directly add the top-down signal onto the bottom-up input of each convolutional layer. Then the pre-tuning and tuning stages are the same as the transformer setting. In our experiments, we choose ConvNeXt (Liu et al., 2022) as the backbone and test on FGVC (Table 9). We observe similar results as in Transformer that TOAST has superior performance than fine-tune and LoRA. This implies attention refocusing is also important for Convnets.

**TOAST is adaptable to larger models.** To see if TOAST can scale to larger models, we test ViT-L pre-trained on ImageNet-21k. As shown in Table 10, on the larger model, TOAST still delivers the best performance. An interesting observation is that LoRA is able to outperform fine-tune in this setting, possibly because the pre-trained representation is strong and general enough and largely modifying the backbone to learn new features is not necessary.

## C  ADDITIONAL RESULTS OF TOAST-LITE

In this section, we provide more results on TOAST-Lite and compare it to other PEFT algorithms such as LoRA and VPT. For visual classification, we show the results on FGVC in Table 11. We can see the TOAST-Lite has a similar number of tunable parameters as LoRA and VPT while obtaining better or comparable performances on all five datasets. It also outperforms fine-tuning while other PEFT methods fail to. We also show the results on VTAB-1k (Table 12). We observe that TOAST-Lite normally is not able to match the performance of TOAST. TOAST-Lite performs on par with LoRA on classification while has a worse performance on structure understanding.

We also provide more results of TOAST-Lite on language generation task. As shown in Table 13, TOAST-Lite has a downgraded performance compared to TOAST, but still outperforms LoRA.

Table 12: Results on VTAB-1K benchmark with TOAST and TOAST-Lite.

| | # Params | Natural | | | | | | | Specialized | | | | Structured | | | | | | | |
|---|---|---|---|---|---|---|---|---|---|---|---|---|---|---|---|---|---|---|---|---|
| | | Cifar100 | Caltech101 | DTD | Flower102 | Pets | SVHN | Sun397 | Camelyon | EuroSAT | Resisc45 | Retinopathy | Clevr-Count | Clevr-Dist | DMLab | KITTI-Dist | dSpr-Loc | dSpr-Ori | sNORB-Azim | sNORB-Ele |
| *ImageNet-1k pre-trained* | | | | | | | | | | | | | | | | | | | | |
| Fine-tune | 87M | 44.7 | 77.3 | 55.5 | 74.5 | 86.0 | **85.1** | 17.4 | **84.9** | 95.0 | 82.8 | 74.2 | 60.2 | 53.1 | 33.5 | 77.6 | 61.9 | 39.0 | 15.0 | 36.6 |
| VPT | 0.9M | 65.3 | 90.5 | 67.7 | 88.3 | 88.6 | 82.2 | 40.6 | 82.3 | 94.5 | 83.1 | 74.0 | 51.5 | 51.1 | 44.1 | 69.3 | 63.8 | 49.5 | 25.3 | 28.6 |
| LoRA | 0.3M | 69.3 | 88.8 | 66.6 | 90.3 | **90.3** | 81.9 | 41.5 | 83.4 | 94.8 | 83.5 | **75.0** | **66.8** | 56.9 | **48.9** | 77.6 | 76.2 | **53.5** | 26.6 | 37.1 |
| TOAST | 14M | **73.8** | **92.1** | **68.7** | 93.0 | 89.0 | 76.3 | 41.9 | 82.8 | 95.3 | 85.7 | 74.6 | 61.2 | **58.7** | 43.5 | **78.8** | **86.1** | 51.2 | **27.0** | **43.4** |
| TOAST-Lite | 0.9M | 69.0 | 91.0 | 65.9 | **93.3** | 88.0 | 78.9 | **42.9** | 81.6 | **96.1** | **86.0** | 74.3 | 62.8 | 53.3 | 44.8 | 73.7 | 66.0 | 41.2 | 26.5 | 39.0 |

Table 13: Results of Instruction Tuning on Alpaca. We use GPT-4 to evaluate the performance. LoRA has inferior performance while TOAST is able to outperform fully fine-tuned Alpaca.

| | # Params | Generic | Knowledge | Roleplay | Common | Fermi | Counterfactual | Writing | *Avg* |
|---|---|---|---|---|---|---|---|---|---|
| Alpaca | 7B | 7.8 | 8.5 | **8.0** | **8.0** | 4.3 | **8.7** | **9.7** | *7.9* |
| LoRA | 4.2M | 6.7 | 7.3 | 6.7 | 7.0 | 5.0 | 7.3 | 8.0 | *6.9* |
| TOAST | 537M | **8.0** | 9.0 | 7.7 | **8.0** | **7.0** | 8.0 | 8.7 | ***8.1*** |
| TOAST-Lite | 19M | 7.3 | **9.3** | 6.7 | 7.8 | 5.1 | 7.0 | 8.3 | *7.4* |

# D    ADDITIONAL RESULTS OF LANGUAGE GENERATION

Figure 7-11 show additional results on language generation for Alpaca-7B, Vicuna-7B, and Vicuna-13B.

> **How can I improve my time management skills?**

**Alpaca-7B:**

Improving time management skills involves breaking down tasks into smaller, more manageable chunks, setting realistic goals and deadlines, and prioritizing tasks based on importance. It also involves being mindful of how much time certain tasks take and scheduling in breaks throughout the day. Additionally, it is important to be aware of distractions and how they can impact productivity. Finally, it is helpful to have a system in place to track progress and hold oneself accountable. With practice and dedication, anyone can improve their time management skills.

**Alpaca-7B-LoRA:**

One of the best ways to improve your time management skills is to create a schedule and stick to it. This will help you prioritize your tasks and ensure that you are spending your time wisely. It is also important to be realistic about how much time each task will take, so that you don't overestimate or underestimate how long it will take you to complete a task. Finally, it is important to learn how to say "no" when you need to, so that you don't take on too many responsibilities at once.

**Alpaca-7B-TOAST:**

One way to improve your time management skills is to create a schedule and stick to it. Make sure to prioritize tasks and set realistic deadlines for yourself. Break down large tasks into smaller, more manageable chunks and take regular breaks to stay focused. Additionally, try to avoid procrastination and distractions by blocking out specific times of the day for certain tasks. Finally, make sure to reward yourself when you complete a task or reach a goal. This will help motivate you to continue working towards your goals.

Another way to improve your time management skills is to practice mindfulness. Mindfulness involves being present in the moment and focusing on the task at hand. This can help you stay focused and reduce distractions. Additionally, mindfulness can help you become more aware of your thoughts and emotions, which can help you make better decisions about how to use your time.

Finally, it's important to remember that time management is a skill that can be improved over time. Practice makes perfect, so don't be discouraged if you don't get it right the first time. With patience and persistence, you will be able to improve your time management skills.

Good luck!

Figure 5: The responses given the question *How can I improve my time management skills*.

**Can you explain the basics of quantum computing?**

**Vicuna-7B:**

Quantum computing is a type of computing that uses the principles of quantum mechanics to perform calculations and process information. It is based on the idea that particles, such as atoms or subatomic particles, can exist in multiple states at once, and can be manipulated and controlled using quantum bits, or qubits.

One of the key principles of quantum computing is superposition, which allows a qubit to exist in multiple states simultaneously. For example, a qubit could represent both a 1 and a 0 at the same time. Another important principle is entanglement, which allows two qubits to be connected in such a way that the state of one qubit affects the state of the other, even if they are separated by a great distance.

Quantum computing has the potential to solve certain problems much faster than classical computers, but it is still a developing field and there are many challenges to overcome before it becomes practical for widespread use.

**Vicuna-7B-TOAST:**

Sure, I'd be happy to explain the basics of quantum computing! Quantum computing is a relatively new field of computer science that uses the principles of quantum physics to perform certain types of computation that are impossible or significantly more difficult to perform on classical computers.

At the most fundamental level, quantum computers are based on quantum bits (qubits) that can exist in a superposition of 0 and 1, allowing them to store and process information in a way that is fundamentally different from classical computers. This superposition and interference phenomena can be used to perform quantum algorithms, which can solve certain problems much faster than on classical computers.

One of the key differences between classical computers and quantum computers is that quantum computers use quantum bits (qubits) that can exist in a superposition of 0 and 1, while classical computers use classical bits (bits) that can only exist in a state of 0 or 1. The ability to store and process information in superposition allows quantum computers to perform certain types of computation that are impossible or significantly more difficult to perform on classical computers.

Quantum algorithms take advantage of the superposition and interference phenomena to solve certain problems faster than on classical computers. One example is Shor's algorithm, which can factor large integers in polynomial time. This algorithm uses quantum computers to search for solutions to a discrete logarithm problem in a large finite field, which is a fundamental problem in cryptography. This algorithm can factor large integers much faster than on classical computers, making it a powerful tool for cryptography.

Another example is Grover's search algorithm, which can search an unsorted database of size N for an item of interest in $O(\sqrt{N})$ time. This algorithm uses quantum computers to search a database with a large number of elements in polynomial time, which is impossible to do on classical computers.

Overall, quantum computing is a relatively new field of computer science that uses the principles of quantum physics to perform certain types of computation that are impossible or significantly more difficult to perform on classical computers. It has the potential to revolutionize the way we think about computation and has applications in a wide range of fields including cryptography, simulation, optimization, and machine learning.

Figure 6: The responses given the question *Can you explain the basics of quantum computing*.

**If you were a Shakespearean character, how would you declare your love for someone in a soliloquy?**

**Vicuna-7B:**

If I were a Shakespearean character, I might declare my love for someone in a soliloquy like this:

"Oh, love, thou art a fickle thing,
A shadow that doth dance upon the wind,
A bird that doth sing thy song in spring,
Yet vanisheth like a fleeting dream.

But oh, how my heart doth yearn
For the one I love, so fair and true,
With eyes like stars that shine so bright,
And a laughter that doth light the night.

Oh, how I long to wrap my arms
Around her slender form, to hold her tight,
To tell her of my love so true,
And to hear her whisper back to me,
"I love you too."

Oh, love, thou art a precious thing,
A treasure that doth shine so bright,
And though I may not always understand,
Thy ways and thy mysteries unraveled,

I know that thou art worth all the pain,
All the heartache and all the sorrow,
For with thee, my heart doth find peace,
And in thy embrace, I'm no longer alone.

So here's to love, oh love so pure,
And to the one I love, dear and true,
I'll spend my days and nights with thee,
And in the end, I'll be with thee."

**Vicuna-7B-TOAST:**

If I were a Shakespearean character, I might declare my love for someone in a soliloquy in one of the following ways:

1. I would declare my love for someone in a soliloquy by saying things like, "O Romeo, Romeo, wherefore art thou Romeo? / All is not well. My heart is sick with passion / And love doth lie too deep for words to sound the mind." (from "Romeo and Juliet")

2. I would declare my love for someone in a soliloquy by saying things like, "O, that I were a glove upon that hand, / That I might touch that cheek, and take those lips / In mine own hand, and press them daintily / For self-love, and give them liberty / To speak what they suppose in me disguised" (from "The Merchant of Venice")

3. I would declare my love for someone in a soliloquy by saying things like, "O mistress mine, where are you roaming? / O stay and hear, your true love's coming, / That can sing both high and low, / While I have music in my ear, / To make you merry, that shall make me sad / When I have music in my ear, / To make me merry, that shall make me sad." (from "Twelfth Night")

4. I would declare my love for someone in a soliloquy by saying things like, "O brave new world that has such people in it / And such people in it! And such people in it!" (from "The Tempest")

5. I would declare my love for someone in a soliloquy by saying things like, "O, that I were a man! / I would fly from this mortal coil / And in my place a spirit should inhabit / And think my thoughts with me, and tell / With what kind of bodies it is furnish'd / And what it thinks, and what it does, and why" (from "Hamlet").

Of course, Shakespeare's plays are filled with memorable soliloquies that express love in many different ways and in many different contexts. My own declaration of love might not be exactly the same as those of these characters, but it would surely be just as heartfelt and meaningful.

Figure 7: The responses given the question *If you were a Shakespearean character, how would you declare your love for someone in a soliloquy.*

> **What if the Aztecs had successfully repelled the Spanish conquistadors?**

**Vicuna-13B:**

It is impossible to know for certain what would have happened if the Aztecs had successfully repelled the Spanish conquistadors, as this is a counterfactual scenario that cannot be verified. However, it is possible to speculate on the potential outcomes based on the available historical information.

If the Aztecs had managed to drive the Spanish out of Mexico, it is likely that the Aztec empire would have continued to exist in some form. The Aztecs were a powerful and sophisticated civilization, with a well-organized government and a strong military. However, it is also possible that the Aztecs would have faced challenges and conflicts from other groups in the region, such as the Maya or the Incas.

Without the influence of the Spanish, it is uncertain how the Aztecs would have developed and evolved over time. It is possible that they would have continued to expand their empire and exert control over neighboring territories, or they may have become more isolated and focused on internal affairs. It is also possible that the Aztecs would have faced challenges and conflicts from other groups, which could have led to their downfall or decline.

Overall, it is difficult to predict exactly how the world would have changed if the Aztecs had successfully repelled the Spanish conquistadors. However, it is clear that the Aztecs were a powerful and influential civilization, and their legacy has left a lasting impact on the region and the world.

**Vicuna-13B-TOAST:**

If the Aztecs had successfully repelled the Spanish conquistadors, the history of Mexico and the Americas would have been extremely different. Here are some of the key ways in which the Aztec Empire's defeat at the hands of the Spanish conquistadors changed the course of history:

1. The Spaniards would not have established a colonial empire in the Americas.

The Spanish conquistadors began a campaign of conquest and colonization in the Americas, which led to the establishment of numerous Spanish settlements and colonies in Mexico, Central America, and South America. If the Aztecs had been able to repel the Spanish, it is likely that the Spanish would not have been able to establish colonies in the Americas, and the history of colonization in the region would have been very different.

2. The development of the Americas would have been slowed down.

The Spanish colonization of the Americas led to a rapid development of the region, as the Spanish brought with them new technologies, agricultural techniques, and social and political institutions. If the Aztecs had been able to repel the Spanish, it is likely that the development of the region would have been slower and more gradual, as the Spanish would not have brought their advanced technologies and institutions to the region.

3. The spread of Christianity in the Americas would have been limited.

The Spanish colonization of the Americas was closely linked to the spread of Christianity in the region. The Spanish colonists brought with them Catholicism, which they attempted to spread among the indigenous populations of the Americas. If the Aztecs had been able to repel the Spanish, it is likely that the spread of Christianity would have been much slower and more limited.

4. The Americas would have been more independent and less integrated into the global economy.

The Spanish colonization of the Americas led to the establishment of a global economy that was tightly linked to Spain and the rest of Europe. If the Aztecs had been able to repel the Spanish, it is likely that the Americas would have developed more independently and less integrated into the global economy.

5. The indigenous population of the Americas would have been more culturally and linguistically diverse.

The Spanish colonization of the Americas led to the widespread adoption of Spanish as the primary language in many parts of the region, and the destruction of many indigenous languages and cultures. If the Aztecs had been able to repel the Spanish, it is likely that the indigenous population of the Americas would have been more culturally and linguistically diverse, with many different indigenous languages and cultures surviving to the present day.

Figure 8: The responses given the question *What if the Aztecs had successfully repelled the Spanish conquistadors*.

> **How can you determine if a restaurant is popular among locals or mainly attracts tourists, and why might this information be useful?**

**Vicuna-13B:**

Determining if a restaurant is popular among locals or mainly attracts tourists can be challenging, but there are a few factors that may provide clues. For example, if a restaurant has a lot of local customers and a limited number of tourists, it may be considered a local favorite. On the other hand, if a restaurant has a high number of tourists and a limited number of local customers, it may be more of a tourist attraction.

This information can be useful for several reasons. If you are a local looking for a new restaurant to try, knowing that a restaurant is popular among locals can give you confidence that it is a good choice. On the other hand, if you are a tourist looking for a local experience, you may want to seek out restaurants that are popular among locals. Additionally, knowing if a restaurant is popular among locals or attracts more tourists can help you plan your itinerary and budget accordingly. For example, you may want to budget more for a restaurant that is popular among locals and attracts a higher price point, or you may want to seek out more affordable tourist attractions.

**Vicuna-13B-TOAST:**

To determine if a restaurant is popular among locals or mainly attracts tourists, you can consider several factors, including:

1. Demographics: Examine the demographics of the surrounding area to see if it is primarily populated by locals or tourists.
2. Language: Listen to the language being spoken by customers and employees to see if it is primarily English or a foreign language.
3. Amenities: Examine the amenities offered by the restaurant to see if they are geared towards locals or tourists. For example, a restaurant that offers local dishes and drinks, and has a relaxed atmosphere may be more popular among locals, while a restaurant that offers international cuisine and has a more formal setting may be more popular among tourists.
4. Prices: Compare the prices of the menu items to other similar restaurants in the area to see if they are comparable to what locals can afford, or if they are more expensive and geared towards tourists.
5. Crowd: Observe the crowd to see if it is primarily made up of locals or tourists. For example, if the crowd is primarily made up of locals who speak the local language, it may be a popular restaurant among locals.

Understanding whether a restaurant is popular among locals or tourists can be useful for several reasons:

1. It can help you determine whether a restaurant is the right fit for you based on your preferences and budget.
2. It can help you identify restaurants that may not be popular with tourists, offering you a more authentic local dining experience.
3. It can help you avoid overcrowded restaurants that may be difficult to get a table at during peak hours.
4. It can help you identify restaurants that may be more likely to offer deals and discounts, as they are more likely to have a larger customer base.
5. It can help you identify restaurants that may be more likely to have a local menu, rather than a tourist-focused menu.

In summary, determining if a restaurant is popular among locals or mainly attracts tourists can help you make informed decisions about where to eat, whether it's for a casual dinner or a special occasion.

Figure 9: The responses given the question *How can you determine if a restaurant is popular among locals or mainly attracts tourists, and why might this information be useful*.

