# OpenReview forum: "TOAST: Transfer Learning via Top-Down Attention Steering"
_ICLR.cc/2024/Conference — ICLR 2024 Conference Withdrawn Submission_

### Official Review · Reviewer_fXJ4 · 2023-11-01

**Soundness:** 3 good
**Presentation:** 3 good
**Contribution:** 2 fair
**Rating:** 3
**Confidence:** 4

**Summary:**

The submission introduces Top-Down Attention Steering (TOAST), a novel transfer learning algorithm designed to improve performance by refocusing model attention on task-relevant features while keeping the pre-trained backbone frozen. TOAST demonstrates state-of-the-art results on various transfer learning benchmarks across both vision and language domains, outperforming established methods like full fine-tuning, LoRA, and prompt tuning. The paper’s key contributions include identifying shortcomings in current transfer learning attention mechanisms, proposing the TOAST algorithm, and showcasing its effectiveness through significant performance improvements on multiple benchmarks.

**Strengths:**

1. **Demonstrated Attention Results**: TOAST’s capability to refocus attention within Transformer-based methods has produced impressive results, illustrating a powerful and explainable feature selection mechanism. This not only enhances the model's performance but also contributes significantly to the transparency and interpretability of the learning process.

2. **Good Writing**: The clarity and coherence in the writing of the submission make complex concepts accessible, facilitating a deeper understanding of TOAST and its innovative approach to transfer learning. The well-structured presentation of ideas ensures that the significant contributions of TOAST are effectively communicated and appreciated by the reader.

**Weaknesses:**

1. **Questionable Motivation**: The authors base their argument on the intuition that noisy attention is detrimental; however, it could be argued that the contrast between the foreground and background could actually aid in recognition. A controversial result in the paper is that fine-tune has the lowest mIoU in Tab.1 while it obtains the best results in Tab.4.

2. **Dependence on Pre-training**: The TOAST method undergoes pre-training using other datasets, but other methods do not include the pre-training process. This raises questions about the fairness of the comparison, the potential time consumption of the pre-training process, and the impact of the pre-training dataset on downstream tasks. Additionally, the application value is questionable if maintaining a pre-training dataset is a necessity.

3. **Time-Consuming**: The method is time-consuming and does not exhibit any clear advantage in terms of efficiency when compared to other methods.

4. **Similar to Top-Down Visual Attention**: The paper does not compare TOAST with the "Top-down visual attention from analysis by synthesis" method, despite the apparent similarities. A clear distinction and rationale for not including this comparison are missing.

5. **Limited Support**: In Fig. 4 and Tab. 1, the attention area highlighted is quite small and, while it may be applicable for bird classification, one might concern if it is a general case. The paper also lacks examples of attention maps in text-based tasks, leaving the reader uncertain about TOAST's performance in different domains.

6. **No Comparison with Newer Fine-Tuning Methods**: The compared methods seem to be out-of-date and therefore reduce the significance of TOAST.

**Questions:**

Please refer to the weaknesses.

---

### Official Review · Reviewer_J2p1 · 2023-11-01

**Soundness:** 2 fair
**Presentation:** 3 good
**Contribution:** 2 fair
**Rating:** 5
**Confidence:** 3

**Summary:**

The authors propose a transfer learning method (TOAST) wherein top-down attention is used to direct the attention to relevant target task features. Specifically, the method involves i) feedforward computation, ii) token weighting through cosine similarity to a learned task-specific latent and feature (channel) selection through a learned linear transform, iii) feedback pathway projections and iv) adding feedback representation to the original 'values' of the self-attention layers.

The method is evaluated on fine-grained object/image recognition and segmentation benchmarks (using Imagenet-ViT as the source task model) as well as language generation (using the LLaMA model family). Since the method weights are randomly initialized, pre-training is first performed using Imagenet for vision benchmarks and OpenWebText for language benchmarks, and then trained for the target task. Experiments show performance benefits over existing methods (namely visual prompt tuning, low-rank adaptation and traditional finetuning) and visualization shows that feedback step makes attention more focused. Finally, method variants are provided with more parameter efficiency (TOAST-Lite) and computational-efficiency (TOAST-Late), and ablations are provided to identify contribution of method components.

**Strengths:**

- The application of top-down computation to 'steer' or direct attention (token and feature selection) is well suited for transfer learning and appropriately described. The initial analysis in table 1 indicates that current transfer learning methods may not attend to target-task features.

- Experiments are performed on multiple established benchmarks for fine-grained classification (FGVC), visual recognition (VTAB-1k) and semantic segmentation (PASCAL VOC and ADE20K) besides language generation datasets (Alpaca and ShareGPT).

- Performance benefits on benchmarks show TOAST performs better or competitively with prompt tuning or transfer learning methods such as visual prompt tuning (VPT) and low-rank adaptation for large-language-models (LoRA), although it requires more compute and parameters (e.g. on FGVC, TOAST is 14M parameters while LoRA and VPT are 0.3M and 0.9M respectively). However, further variants with lower computation and parameters perform competitively (e.g. TOAST-Lite which uses LoRA has 0.9M params and still maintains an average improvement over baselines).

- Results with LLaMA models for language generation benchmarks show applicability of method for both vision and language tasks.

**Weaknesses:**

- The novelty of the method appears limited. The top-down attention method proposed is in design and intuition very similar to recent works [1, 2] in top-down attention for computer vision tasks. While [1] has been referenced, the core top-down method (utilizing cosine similarity with a latent, and a feedback pathway) is very similar to that work being applied for transfer learning. [2] has also studied top-down attention in adapting vision backbones using few parameters for vision tasks including fine-grained classification, and has similarly proposed usage of spatial and feature attention for task-relevant feature selection. The differentiation of TOAST from these existing works can be improved and made more explicit in writing to highlight the contribution/novelty of this work.

- It is unclear (or a possible error) how average is calculated for table 3 (results on VTAB-1K benchmark). E.g. average of accuracies for ImageNet-1k pretrained TOAST is reported as 72.5% but when individual accuracies (73.8 92.1 68.7 93.0 89.0 76.3 41.9 82.8 95.3 85.7 74.6 61.2 58.7 43.5 78.8 86.1 51.2 27.0 43.4) are averaged it comes to 69.6%. This is also true for other results/baselines in table 3 (all appear to have lower mean accuracy than reported), so it is perhaps a minor/unintended error or a different averaging method is used.

- It is not indicated which variant of VPT [3] (VPT-Shallow or VPT-Deep) is used for comparison. Ideally, comparisons should have been performed with both variants since both have lesser parameters and computation than the TOAST method and don't require pre-tuning. For VPT-Deep ImageNet-21K finetuned, the reported average performance by VPT paper [3] (table1) on VTAB-1K is 69.4% which is close to TOAST mean average of 69.6%. Similarly, the reported average acc. on FGVC for VPT-Deep VIT-B/16 ImageNet-21k by VPT-paper [3] (table 1) is 89.1% which is higher than TOAST-ViT-L ImageNet-21k average acc. of 88.4% (although the VIT model differs in this case).

- Computational FLOPs comparison against FGVC baselines (VPT and LoRA) is not provided, making it unclear how much additional compute is required compared to these methods. Further, it is unclear (or a possible error) why Table 6 results for LoRA and VPT (79.8% and 78.0% respectively) on FGVC are different from their results in table 2 (81.8% and 82.6%) since their does not appear to be any change to these methods when comparing with TOAST-Lite.


[1] Shi, B., Darrell, T., & Wang, X. (2023). Top-Down Visual Attention from Analysis by Synthesis. In Proceedings of the IEEE/CVF Conference on Computer Vision and Pattern Recognition (pp. 2102-2112).

[2] Jaiswal, S., Fernando, B., & Tan, C. (2022, October). TDAM: Top-down attention module for contextually guided feature selection in cnns. In European Conference on Computer Vision (pp. 259-276). Cham: Springer Nature Switzerland.

[3] Jia, M., Tang, L., Chen, B. C., Cardie, C., Belongie, S., Hariharan, B., & Lim, S. N. (2022, October). Visual prompt tuning. In European Conference on Computer Vision (pp. 709-727). Cham: Springer Nature Switzerland.

**Questions:**

Please refer to weaknesses section.

---

### Official Review · Reviewer_m6yv · 2023-11-03

**Soundness:** 2 fair
**Presentation:** 3 good
**Contribution:** 2 fair
**Rating:** 5
**Confidence:** 4

**Summary:**

This paper investigates the problem of transferring a pre-trained model to novel downstream tasks.  This paper gives a new perspective of attention towards task-relevant features. By refocusing model attention on task-specific parts, they introduce Top-Down Attention Steering (TOAST). With experiments on a range of fine-grained visual classification datasets and instruction-following language generation tasks,  they verify the effectiveness of TOAST and its superiority over fully fine-tuning, LoRA, and prompt tuning.

**Strengths:**

1. The proposed method is simple and effective without introducing extra hyperparameters
2. The experiments are extensive, covering a wide range of applications including both computer vision and natural language processing.
3. The paper is well-organized and easy to follow, with a good visualization of attention maps.

**Weaknesses:**

1. The proposed method is not that strong. According to the ablation study in Table 8, without pre-tuning, TOAST is even lower than VPT.  Further, the largest improvement also comes from the pre-tuning strategy. However, pre-tuning the top-down attention on a general public dataset such as ImageNet or OpenWebText to get a better initialization is not always available and needs to introduce extra datasets. For example, it may be difficult to pre-tune on ImageNet for a downstream task of medical imaging, since the former has natural images while the latter focuses on human tissues or organs.
2. As mentioned by the authors, one drawback of TOAST is the computation overhead since the feedforward path is run twice, which approximately doubles the FLOPs of the model. According to the results in Table, even with the computation-efficient variant (TOAST-Late), the proposed method still needs much more FLOPs than Fine-tune (1.4x v.s. 1.0x in both FGVC and Vicuna).
3. As for the novelty of the proposed method, I found it quite similar to SENet [1], which also enables channel attention with MLP.

[1] Squeeze-and-Excitation Networks.

**Questions:**

1. The authors mentioned that each attention map is averaged across different heads in the last layer of ViT.  I am curious about the motivation for choosing such a mechanism to generate an attention map. What about using Class Activation Mapping (CAM) [1]
2.  In the evaluation of language generation, the authors use GPT-4 (OpenAI, 2023) to score the answers provided by the model. Will this strategy be biased on the GPT-4? Is it a proper evaluation strategy?


[1] Zhou et al. “Learning Deep Features for Discriminative Localization.” CVPR 2016

**Details Of Ethics Concerns:**

No ethics review is needed.